# Sequence-structure-function relationships in the microbial protein universe

Julia Koehler Leman [1,2,19] ✉, Pawel Szczerbiak[3,19], P. Douglas Renfrew[1,2,19], Vladimir Gligorijevic[1,4], Daniel Berenberg[1,4,5,6], Tommi Vatanen[7,8,9], Bryn C. Taylor[10,16], Chris Chandler[1], Stefan Janssen[11,17], Andras Pataki[12], Nick Carriero[12], Ian Fisk[12], Ramnik J. Xavier [7,13], Rob Knight [10,11,14,15], Richard Bonneau[1,2,5,6,18] & Tomasz Kosciolek [3,19] ✉

For the past half-century, structural biologists relied on the notion that similar protein sequences give rise to similar structures and functions. While this assumption has driven research to explore certain parts of the protein universe, it disregards spaces that don't rely on this assumption. Here we explore areas of the protein universe where similar protein functions can be achieved by different sequences and different structures. We predict ~200,000 structures for diverse protein sequences from 1,003 representative genomes across the microbial tree of life and annotate them functionally on a per-residue basis. Structure prediction is accomplished using the World Community Grid, a large-scale citizen science initiative. The resulting database of structural models is complementary to the AlphaFold database, with regards to domains of life as well as sequence diversity and sequence length. We identify 148 novel folds and describe examples where we map specific functions to structural motifs. We also show that the structural space is continuous and largely saturated, highlighting the need for a shift in focus across all branches of biology, from obtaining structures to putting them into context and from sequence-based to sequence-structure-function based meta-omics analyses.

Structural biology follows the sequence-structure-function paradigm, which states that the sequence of a protein determines its structure, which in turn, determines its function[1–4]. Experimental structure determination efforts were unable to keep up with the exponential growth of available sequences, yet recent breakthroughs in protein structure prediction and renewed focus on machine learning approaches, through methods like AlphaFold2[5], now allow for closing the sequence-structure gap. While disordered sequences, large

[1]Center for Computational Biology, Flatiron Institute, Simons Foundation, New York, NY, USA. [2]Department of Biology, New York University, New York, NY, USA. [3]Malopolska Centre of Biotechnology, Jagiellonian University, Krakow, Poland. [4]Prescient Design, a Genentech accelerator, New York, NY 10010, USA. [5]Center for Data Science, New York University, New York, NY 10011, USA. [6]Courant Institute of Mathematical Sciences, Department of Computer Science, New York University, New York, NY, USA. [7]Broad Institute, Cambridge, MA, USA. [8]Liggins Institute, University of Auckland, Auckland, New Zealand. [9]Research Program for Clinical and Molecular Metabolism, Faculty of Medicine, 00014 University of Helsinki, Helsinki, Finland. [10]Department of Pediatrics, University of California San Diego, La Jolla, CA, USA. [11]Center for Microbiome Innovation, University of California, San Diego, La Jolla, CA 92093, USA. [12]Scientific Computing Core, Flatiron Institute, Simons Foundation, New York, NY, USA. [13]Center for Microbiome Informatics and Therapeutics, MIT, Cambridge, MA 02139, USA. [14]Department of Computer Science and Engineering, University of California San Diego, La Jolla, CA, USA. [15]Department of Bioengineering, University of California, San Diego, USA. [16]Present address: In Silico Discovery and External Innovation, Janssen Research and Development, San Diego, CA 92122, USA. [17]Present address: Algorithmic Bioinformatics, Justus Liebig University Giessen, Giessen, Germany. [18]Present address: Prescient Design, a Genentech accelerator, New York, NY 10010, USA. [19]These authors contributed equally: Julia Koehler Leman, Pawel Szczerbiak, P. Douglas Renfrew, Tomasz Kosciolek. ✉e-mail: julia.koehler.leman@gmail.com; tomasz.kosciolek@uj.edu.pl

complexes, multiple chains, and protein–protein interactions remain to be addressed, the large number of available protein structures and models has drastically shifted the perspective in the field.

Here, we predict the structures of ~200,000 metagenomic sequences leveraging a citizen-science approach. We annotate these models in terms of protein function[6], specifically providing residue-specific annotations, and analyze the features of the resulting protein structure-function universe, including fold novelty and structure-function relationships. Our work demonstrates how to integrate massive structural datasets into a sequence and function context and motivates a shift in perspective to include structurally informed functional annotations as the starting point to understand biological questions.

## Results and discussion

Recent advances in the availability of predicted protein structures, including the AlphaFold database and the MIP database presented here, change the view on protein sequence-structure-function relationships from a relative paucity of structural information to a relative abundance of it. This puts us in a position to start answering fundamental questions previously out of reach. How much of the protein structure and fold space is still unexplored? And can we learn anything new about the sequence-structure-function universe of microbial proteins? Here, we try to answer some of these and other questions by large-scale structure prediction efforts that we relate to the sequence space and residue-specific function prediction.

### A database of 200,000 microbial sequences, structures, and functions

Here we performed large-scale structure prediction on representative protein domains from the Genomic Encyclopedia of Bacteria and Archaea (GEBA1003) reference genome database across the microbial tree of life[7]. A summary of our workflow is shown in Fig. 1a (see also Supplementary Methods section 1). From a non-redundant GEBA1003

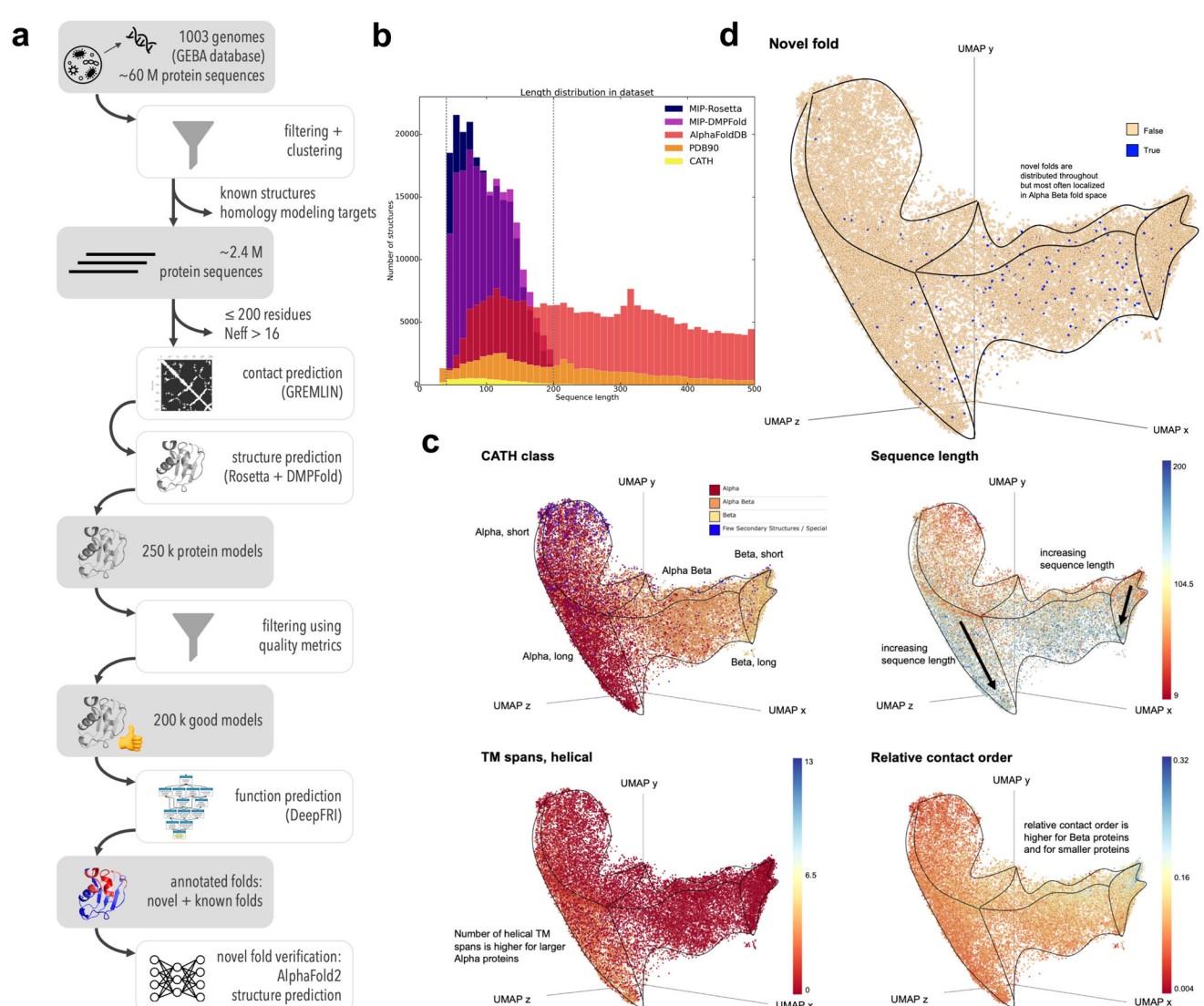

**Fig. 1 | The fold space covered by the microbial protein structure universe is continuous. a** Flowchart of our process to arrive at ~200,000 de novo protein models covering a diverse sequence space. **b** The sequence length distribution shows that our sequences are shorter than many of the proteins in the PDB, CATH or AlphaFold databases, as expected. We predicted structures between 40 and 200 residues long, which covers the majority of length distributions in microbial proteins, which are often shorter than eukaryotic sequences. **c** The protein structure universe in UMAP space is color-coded according to features, such as similarity to CATH classes, sequence length, number of helical transmembrane spans, and relative contact order. **d** Novel folds (blue dots) are spread throughout the fold space with fewer representatives in the purely α-helical and purely β-sheet folds. [Icons in panel (**a**) were created by Ronald Vermeijs and Maxim Kulikov for the Noun Project, licensed under the Creative Common license CCBY3.0]. Source data for this figure are provided in the source data file.

gene catalog we extracted protein sequences without matches to any structural databases and which produced multiple-sequence alignments deep enough for robust structure predictions using Rosetta[8] or DMPfold[9] (N_eff > 16, see Supplementary Methods section 2). For computational tractability we prioritized sequences according to their length and exhaustively sampled all putative novel domains between 40 and 200 residues. For each sequence we generated 20,000 Rosetta de novo models[8] using World Community Grid (formerly IBM) via the Microbiome Immunity Project and up to 5 models per sequence using DMPfold[9]. Unless otherwise stated, we use Rosetta models for the figures in this manuscript. We then curated the initial output dataset (*MIP_raw*) of about 240,000 models to arrive at high-quality models comprising about 75% of the original dataset (*MIP_curated*)—see the next section for details. All analyses in this paper are either on *MIP_curated* or a subset. Functional annotations of the entire dataset were created using structure-based Graph Convolutional Network embeddings from DeepFRI[6]. A detailed description on how to interpret DeepFRI scores and output is provided in Supplementary Methods section 3.

## Model quality assessment metrics to filter out low-quality models

Model quality assessment metrics were derived from 5000 randomly selected proteins (a.k.a. *MIP_random5000_raw* - see Supplementary Note 2) in three steps. First, we noticed that the Rosetta models in our MIP database generally contain fewer coil residues than the DMPfold models (Supplementary Note 3.2), yet the quality of the DMPfold models is higher for larger proteins (see Supplementary Note 3.3.). We therefore filtered by coil content with varying thresholds for the two methods: Rosetta models with >60% coil content, and DMPFold models with >80% coil content were filtered out as these have low quality.

Second, each modeling method needed a quality metric to evaluate the model quality. DMPfold outputs a confidence metric for each model that we used as-is. Rosetta's energy score is only meaningful in relation to other models of the same protein within the energy landscape. For Rosetta, we derived a model quality assessment (MQA) score by averaging the pairwise TM-scores[10] of the 10 lowest-scoring models. If these models sample a minimum in the energy landscape, they are structurally very similar and their average TM-score is high. If Rosetta's scorefunction is unable to identify a specific fold, the average TM-score is low. We filtered out models with an MQA score ≤ 0.4 as these models have low quality.

The third quality metric we use is the agreement between the Rosetta and DMPfold models. If both models are similar (TM-score ≥ 0.5), then we can be confident that they are of high quality. This is supported by the correlation between MQA scores and TM-scores (Supplementary Note 3.3, Supplementary Figure 10). Further, the predictions between Rosetta and DMPfold mostly agree (mean TM-score = 0.61; median TM-score = 0.56)—Supplementary Note 5.3. The quality metrics we used are independent of target difficulty that is often used to classify targets in CASP. Further, all targets in our dataset are "hard" targets since they have low to very low homology to any known structures.

We then used these quality metrics to filter out low quality models: from >240,000 models in the *MIP_raw* database, we arrived at >200,000 models in the *MIP_curated* database (see Supplementary Note 2).

## Identification and verification of novel folds

Putative new folds were identified by comparing our models against representative domains in CATH[11] and the PDB, using a TM-score cutoff[12,13] of 0.5. The output set contained 452 novel structures grouped into 161 fold clusters (Supplementary Information and Supplementary Data 1, and 2). Putative novel folds were also verified by AlphaFold2, which identified 14 false positives, decreasing the number of novel structures to 438, clustered into 148 novel folds.

Supplementary Note 5 describes in detail how novel folds were identified, why specific cutoffs were used and shows false positive clusters.

The agreement between AlphaFold2 and Rosetta or DMPFold is even higher than between Rosetta and DMPFold (see Supplementary Figure 41). We speculate that AlphaFold2 might link physical and knowledge-based scorefunctions (Rosetta) and machine learning approaches (DMPFold) better. We decided not to run AlphaFold2 on the entire MIP dataset due to runtime demands and us questioning whether this new data would provide much insight, as most of our Rosetta and DMPFold models have a high agreement and therefore high confidence.

## The MIP database is orthogonal to existing databases

We wanted to answer the question how similar or different the MIP database is compared to other protein structure databases. The baseline is the PDB90, which are proteins from the Protein Data Bank with a pairwise sequence identity ≤ 90%. CATH superfamilies are a non-redundant subset of the PDB90, covering over 6000 folds (v4.3.0). The AlphaFold protein structure database[5,14] contains over 200 million protein models, vastly increasing the known structure space, and covers a wide range of organisms and sequence lengths, primarily from Eukaryotes. Our MIP database is distinct from the other databases because it consists of proteins from Archaea and Bacteria, whose protein sequences are generally shorter than Eukaryotic[15,16] ones. The average structural domain size for microbial proteins is about 100 residues (Figure 8 in ref. [17]). We predicted structures in the size range from 40 to 200 residues because when we started this project in 2016, structure prediction methods performed better on smaller proteins, 200 residues still cover the majority of length space for microbial proteins, and we wanted to focus on single domains and longer proteins are more likely to cover multiple domains.

MIP models drastically increase the available structure space of smaller proteins and domains from 40 to 200 residues (Fig. 1b), as we selected. We further split the sequences into domains before structure prediction, unlike structures in the AlphaFold database. Also, only about 3.6% of structures in the AlphaFold database belong to Archaea and Bacteria, indicating that AlphaFold and MIP databases are complementary.

## The microbial protein universe maps into a continuous fold space

We wanted to contextualize the MIP dataset in relation to existing structures and to investigate the features of a more complete and less biased protein structure universe[18–21]. The PDB is biased by proteins that are more amenable to structure determination and by proteins of higher interest as pharmaceutical targets, resulting in a larger number of very similar structures with different mutations, ligands and chemical environments.

To generate the visualization, we represent each protein structure as a graph given by its C-alpha contact map below a 6Å threshold. We preprocess each graph by computing a 42-dimensional graphlet vector representation[22,23]. A collection of graphlets up to size K is a set of all possible non-isomorphic induced at most K-sized subgraphs of a given graph G. Graphlet count vectors[24] report the counts of a set of (computationally tractable) graphlets up to a given size; they serve as a powerful baseline for graph encoding methods that do not consider node level features[25]. For each model in the visualization dataset and CATH superfamilies, we mapped the 42-dimensional graphlet count vectors into 3D space using UMAP dimensionality reduction (Fig. 1c, d). Visualization was done in Emperor[26]. The surfaces of the 3D structure cloud are outlined in black. We investigated several features in this mapping, including sequence length, relative contact order, number of transmembrane spans, and mapping to a CATH class. The 3D mapping of the protein universe allows to distinguish different sequence lengths, the number of helical transmembrane spans and the relative

contact order of the protein folds, as different shadings show in Fig. 1c. The visualization further illustrates (see Supplementary Note 6) that the protein universe space is continuous, indicating that folds may evolve along a trajectory where small changes in the tertiary structure can eventually lead to a different fold. Our results are in agreement with prior work, albeit derived from a different (microbial proteins), larger and more diverse dataset and using a different methodology[27–29]. In contrast, a discrete fold space would display distinct clusters of folds that require larger conformational changes to interconvert between them. Prior work suggested evolutionary or geometric sources for the continuity of the protein fold space, and that a low-dimensional representation has the potential to aid protein structure-function investigations[21,29–31]. We identify 438 previously unseen structures in our MIP dataset that cluster into 148 distinct, novel folds (46 clusters with multiple proteins and 102 singletons—see Supplementary Note 5). Figure 1d shows that the majority of novel folds are distributed throughout α/β fold space (compare with Fig. 1c) with few novel folds in α or β fold space.

### MIP dataset explores the sequence-structure-function universe

To investigate the sequence-structure-function relationships in this microbial protein universe and possibly gain novel insights from less explored corners of this universe, we computed pairwise similarities between random sequences in a subset of the curated dataset (*MIP_random5000_curated*—see Table S2) in terms of sequence identity, structural similarity (TM-score) and functional similarity (cosine similarity score of DeepFRI function prediction vectors - see Supplementary Methods section 3). This was compared against a PDB baseline of 1000 protein chains, covering pairwise sequence similarities between 0 and 100%.

By design, the full-length protein sequences in the MIP dataset are dissimilar (30% sequence identity cutoff) yet pairwise sequence identities between domains can occasionally be higher than 30% (Fig. 2). When correlating sequence identities to structural similarities for pairs of proteins, the vast majority of dissimilar sequences fold into different structures, and a trend can be identified (Supplementary Note 7.1) that has been previously described[32,33]. However, there are a fair number of proteins that have vastly different sequences and still fold into similar structures (Fig. 2b). The PDB baseline that covers sequence similarities across all ranges from 0 to 100% confirms this expected trend, and it also confirms the general notion that similar sequences fold into similar structures (Fig. 2a).

When correlating sequence identity and functional similarity, the majority of sequences have different functions, but still a fair number of dissimilar sequences have similar functions. This originates in the multiplicity of biological systems (Fig. 2d). i.e., achieving the same functional outcome by different pathways (for example[34,35]). The PDB baseline gives the same trend and has an additional known population where similar sequences achieve similar functions (Fig. 2c).

When correlating structural similarity (of dissimilar sequences) to functional similarity, we find 4 populations (Fig. 2f): (a) the largest population following expectations of dissimilar structures having different functions—quadrant III, (b) the 2nd largest population of dissimilar structures having similar functions—quadrant I, (c) the third largest population of similar structures having different functions—quadrant IV, and (d) the smallest population following expectations of similar structures having similar functions. Quadrants I and IV are the most interesting ones with examples shown below. The PDB baseline covers all sequence similarities and follows mainly known expectations of quadrants II and III (Fig. 2e).

### Most functions are produced by the same structural motifs, even for dissimilar sequences

For each of the 148 novel fold structural clusters, we compared functional similarities for each protein pair by computing the cosine

similarity for the predicted function vectors from DeepFRI; this is shown as a heatmap in Fig. 3 (a detailed description on how to interpret DeepFRI scores and output is provided in Supplementary Methods section 3). We then picked several proteins for each structural cluster and mapped selected top-scoring functions onto the predicted structures (lower panel in each subpanel in Fig. 3). Residues that DeepFRI predicts to have high importance to achieve a particular function are highlighted in red, whereas blue residues are not involved in generating that particular function. We find that the majority of functions in those structural clusters map to the same residues in the structure ("structural motif") as shown for the clusters 158 and 153 in Fig. 3a, b (more details for those examples are shown in Supplementary Note 8). However, we also find more complicated structure-function relationships in these clusters as shown in Fig. 3c–e and discussed in the next section. Note that the sequences in each structural cluster (and in the MIP dataset) are dissimilar to each other and neither structural nor functional prediction could be inferred by sequence identity for these proteins due to lack of homology. More examples of this analysis are outlined in Supplementary Data 3 and 4.

### Per-residue functional annotations reveal a more complex picture of protein structure-function relationships

Some of the structure-function relationships map to quadrants I and IV in Fig. 2f, where similar structures can have different functions[36] or different structures can have similar functions. Different structures can generate similar functions due to the multiplicity of functional pathways[34,35] as a back-up plan for organisms to survive. However, a closer look at some of the structure-function disparities reveals some surprises.

Figure 3a, b shows two proteins that use the same structural motif for different functions. While the overall sequence identity between these proteins is low (~30 and 25% for panels A and B, respectively), a short sequence motif underlies the structural motif, which in turn has different functions. Figure 3a shows two proteins where the terminus of the central helix is involved in phosphatase activity, where the same motif in a different protein is involved in actin binding. The sequence motif for this region is GGWDXP. In Fig. 3b, the N-terminus of one protein is involved in zinc ion binding and 'catalytic activity, acting on a protein', whereas the N-terminus of another protein of that structural cluster is involved in DNA binding and 'identical protein binding'. The underlying sequence motif for this structural motif is CXCCG.

Figure 3c–e shows examples where a different structural motif in the same protein fold achieves the same function. This seems unusual and doesn't seem to rely on a short sequence motif. In the first example (Fig. 3c), transferase activity either maps to a beta-sheet or a C-terminal short helix in two different proteins. In the second example transmembrane transporter activity maps to either two helices or a beta-sheet (Fig. 3d). In the third example, ion transmembrane transporter activity maps to different structural motifs for different proteins (Fig. 3e). This entire structural cluster (cluster 146) has very high similarity across predicted functions, indicated by the heatmap showing mostly yellow hues.

### Higher functional specificity is carried out by fewer possible folds

We investigated different protein functions and examined the structures with these functions (Fig. 4 and Supplementary Data 4). Some protein functions are sufficiently general such that they can be achieved by different folds, examples are 'carbohydrate binding' (GO:0030246), 'protein tyrosine kinases' (EC 2.7.10.), and 'mitigation of host immune response by virus' (GO:0030683). More specific functions are accomplished by fewer folds. Examples of specific functions with a single fold in our MIP dataset are 'thymidine kinase' (EC 2.7.1.21) and 'sole sub-class for lyases that do not belong in the other subclasses' (EC 4.99.1.).

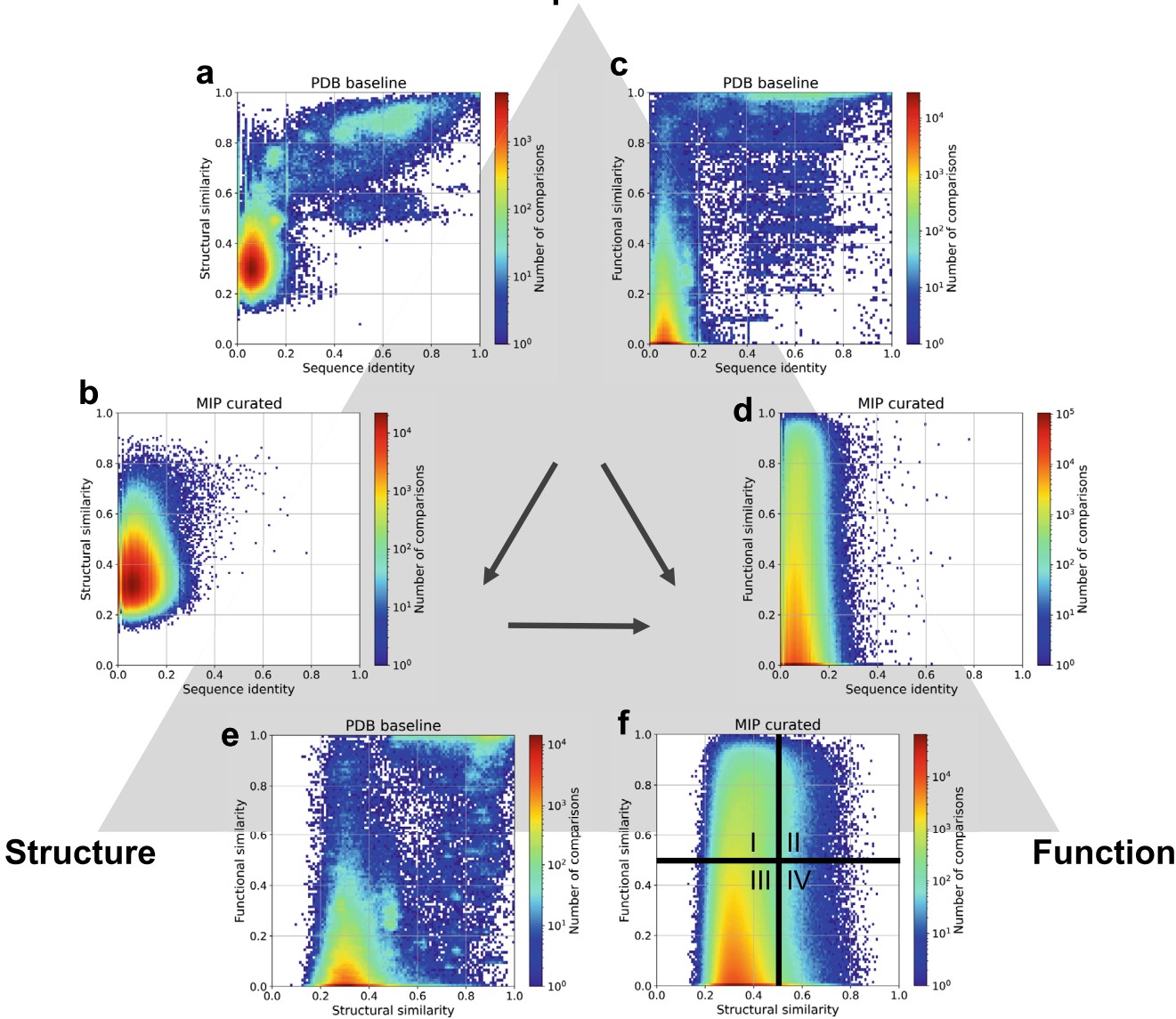

**Fig. 2 | Sequence-structure-function relationships in both PDB and the MIP dataset.** Pairwise comparisons of protein sequences (using sequence identity), structures (TM-score), and functions (cosine similarity between DeepFRI output vectors) for two datasets: a baseline from the PDB and the *MIP_random5000_cu-rated* dataset, containing 3,052 Rosetta generated models (see Tables S2 and S3). The PDB baseline dataset contains 1000 chains covering pairwise sequence simi-larities between 0 and 100% while the MIP dataset is a non-redundant set with mostly dissimilar sequences (sequence identity <30% threshold was imposed before sequential domain splitting). Analyses of these two datasets in this way lead us to the following conclusions: sequence identity and structural similarity follow a

known trend (Supplementary Fig. 72) (**a**), yet high structural similarity can be achieved by low sequence identity (**b**). High sequence identity (sequence identity > 50%) leads to high functional similarity (cosine similarity > 0.5) (**c**), yet high functional similarity can be achieved by proteins with low sequence identity (**d**). Structural similarity often correlates with functional similarity ((**e**) and quadrants II and III in (**f**)). However, there are plenty of examples where low structural similarity can be seen in proteins with high functional similarity (quadrant I in (**f**)), and highly similar structures can exhibit different functions (quadrant IV in (**f**)). Source data for this figure are provided in the source data file.

The functional cluster for carbohydrate binding (Fig. 4a) covers many different folds with high β-sheet propensity, including β-barrels, twisted sheets, and stacked sheets. This class contains a single helical protein, indicated by the single blue line in the heatmap in Fig. 4a with the structure shown in (F). The largest structural cluster in this func-tional category corresponds to the largest novel-fold cluster (yellow square in the heatmap) and the salient residues in this cluster show a high degree of overlap.

Figure 4b shows the function 'maintenance of CRISPR repeat elements'. CRISPR repeats are short DNA sequences in bacteria and archaea. They derive from DNA fragments of bacteriophages that previously infected those organisms and allows them to identify

recurring invaders. Hence, the CRISPR-Cas system functions like a microbial immune system[37]. Cas1 and Cas2 identify the site in the bacterial genome where viral DNA is inserted and ultimately cleaved by Cas9[38]. The structural cluster in Fig. 4b (A) overlays with part of Cas2 (PDB ID 5sd5 or 5xvp[39], chains EF) and the predicted salient residues bind DNA in the structure. Cluster (D) in Fig. 4b is similar to parts of Cas1 (PDB ID 5sd5 or 5xvp[39], chains ABCD) but doesn't overlay perfectly.

Figure 4c shows the function 'sole sub-class for lyases that do not belong in the other subclasses'.

Lyases are enzymes that catalyze the breaking of chemical bonds by means other than hydrolysis or oxidation. None of the lyases in the

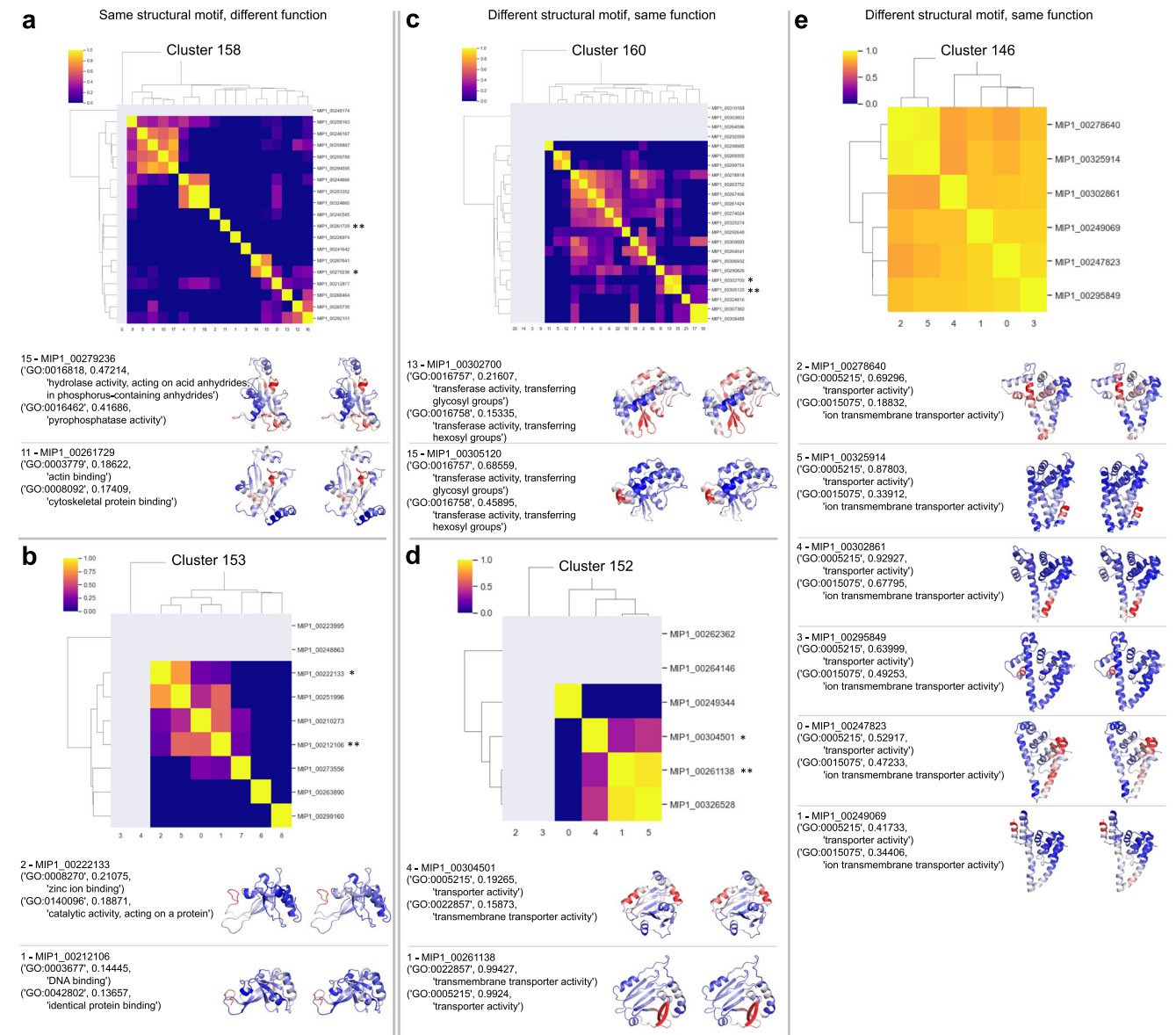

**Fig. 3 | Functional diversity of proteins with the same structure.** We show examples from several structural clusters (Rosetta models) that exhibit novel folds. The heatmaps show functional similarity (cosine similarity of the function vectors) of protein pairs within the cluster. Proteins that have predicted functions with scores <0.1 are shown in gray in the heatmaps. Asterisks highlight the examples shown below. **a, b** Cases where the same structural motif in two different proteins produces different, unrelated functions. **c–e** Cases where the same function is generated by different structural motifs in different proteins, even though the proteins have the same fold. Source data for this figure are provided in the source data file.

other classes (EC. 4.1–EC.4.6) have the same fold as our predicted MIP models, even though there are structural similarities. Our models have an (αβ)×3 fold with sequential strand connections–the other lyases have various (αβ)xN folds but their strand connections are non-sequential.

In summary, in this study, we used a citizen-science approach to predict ~200,000 protein structures for non-redundant microbial sequences across the tree of life. Structures were predicted by two state-of-the-art independent methods (Rosetta and DMPfold) and evaluated by quality metrics to indicate model quality. Functional annotations give us a unique look at the microbial protein universe in terms of sequence, structure, and function. Our database is orthogonal to the AlphaFold database in terms of domains of life, sequence diversity and sequence length. We predicted 148 novel folds which were verified by AlphaFold. With functional annotations, we can more closely relate sequence-structure-function relationships in this universe, that go beyond the main homology assumption of similar sequences, structures, and functions. In fact, we frequently see that these dissimilar sequences fold into similar structures, indicating that the sequence diversity is much greater than the structural diversity. From a structure prediction standpoint, this highlights the importance of distant homology detection and fold recognition methods for dissimilar sequences. Moreover, we provide examples that challenge our classic understanding of biological sequence-structure-function relationships. We hope that this research inspires the scientific community to advance our understanding of site-specific protein function by developing experimental and computational tools for high-quality measurements and predictions. Only these new tools can lead to a more complete understanding of the complexities of how proteins fold, function, evolve and interact, to address questions related to health, disease, and engineering applications to solve some of the world's biggest problems.

**a** MF GO:0030246 - carbohydrate binding

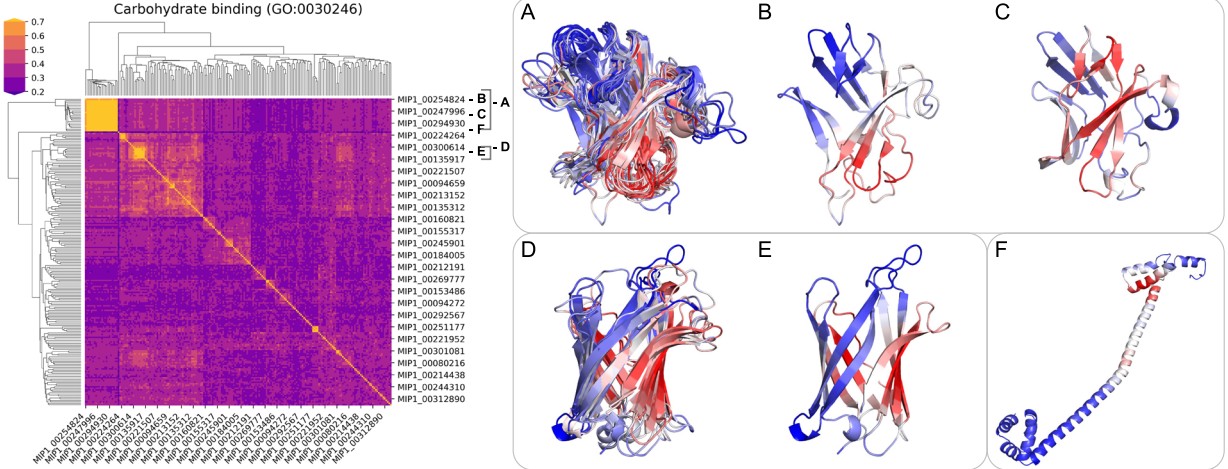

**b** BP GO:0043571 - maintenance of CRISPR repeat elements

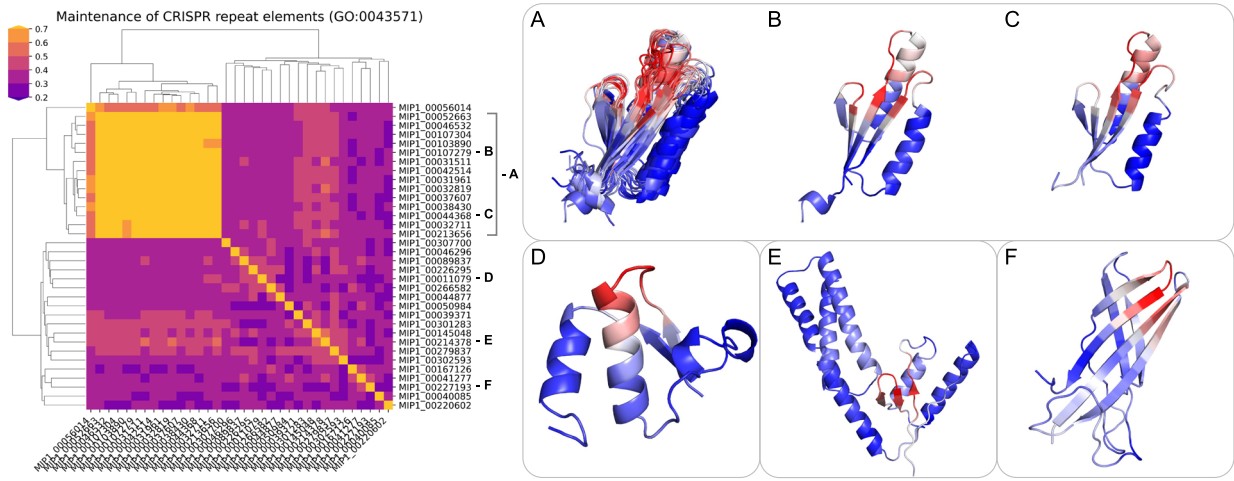

**c** EC 4.99.1.- Sole sub-subclass for lyases that do not belong in the other subclasses

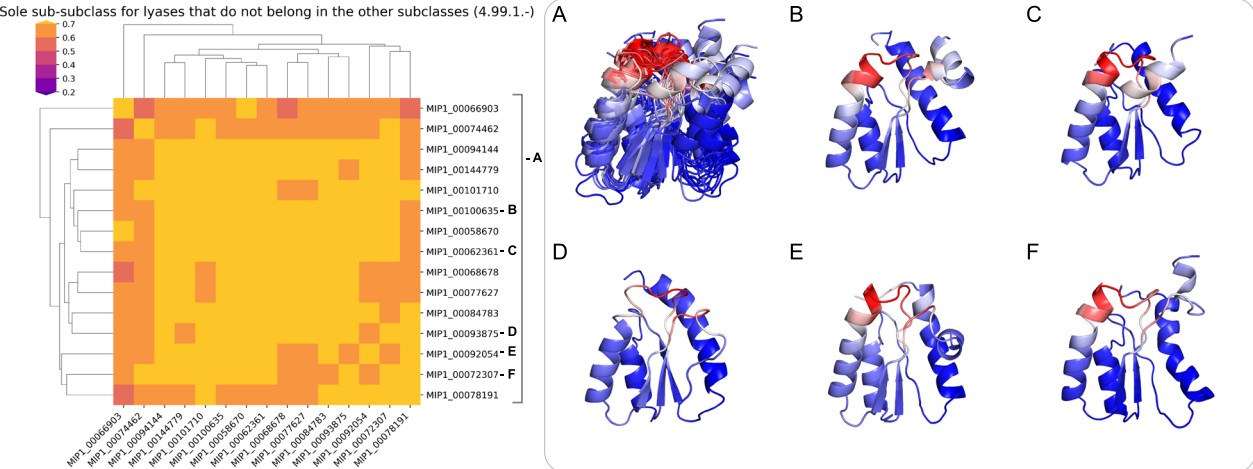

## Methods

### Sequence clustering of GEBA dataset

The MIP dataset is constructed on the basis of GEBA1003 representative bacterial and archeal genomes from across the tree of life[7]. The dataset includes environmental samples from soil, ocean water, human gut microbiome and was designed to sample the microbial tree of life evenly. For each genome, we generated a list of predicted genes using Prodigal[40]. The raw gene catalog was processed using an incremental clustering approach, similar to the one employed by UniClust[41]. First, redundancy in the dataset was removed by using linclust (i.e., clustering at 100% sequence identity), as implemented in MMSeqs2[42,43]. Then, the dataset was further clustered into 90, 70 and

**Fig. 4 | Structural diversity of proteins with the same function.** We examine proteins that have the same function and plot the TM-score as a measure of structural similarity as a heatmap, with larger numbers (more yellow) representing more similar structures. We also map the residue-specific function predictions onto the structures on the right, where residues in red are responsible for the functions. **a** Gene ontology molecular function carbohydrate binding with GO number GO:0030246. Except for the protein shown in (F) which has high helical propensity, the proteins in this functional cluster have high β-sheet content. The largest cluster in the heatmap in yellow is also the largest novel-fold cluster. The salient residues

responsible for this function overlay nicely across the proteins in this cluster. **b** Gene ontology biological process function 'maintenance of CRISPR repeat elements' with GO number GO:0043571. The largest cluster highlighted in yellow superimposes with Cas2 and the salient residues in red interact with DNA. **c** Enzyme commission number EC 4.99.1. with the function 'Sole sub-class for lyases that do not belong in the other subclasses'. All structures in this functional cluster have the same fold and the salient residues responsible for this function overlay onto the same structural motif in the protein. More details in the text. Source data for this figure are provided in the source data file.

30% sequence identity clusters, with the last step (70–30% clustering) executed using the MMSeqs2 `clust` module. The resulting dataset was sorted according to sequence length, sampling the entirety of sequences between 40 and 200 residues.

## Structure prediction

AlphaFold wasn't available for the majority of the life of this project, which started in 2016, and computational runtime and initial lack of code availability prevented us from running it on the World Community Grid. We used Rosetta and DMPFold for protein structure prediction, two state-of-the-art methods available at the time.

**Rosetta structure prediction.** The structure of each MIP sequence was predicted using version 2016.32.58837 of the Rosetta Macromolecular Modeling Suite, modified to run on the IBM World Community Grid. Residue-residue contacts from sequences closely related to the target sequence were inferred using GREMLIN[44] (version 2.0.1) and incorporated as constraints during the folding protocol. For each MIP sequence, 20,000 models were generated. Models were ranked using the REF2015 energy function[45] and the lowest energy model was used for further analysis. Details of the fragment selection, contact prediction, and Rosetta model building can be found in the supplement.

**DMPfold structure prediction.** We additionally predicted the structures of all MIP sequences using DMPFold[9]. The same multiple sequence alignments used for contact prediction in the Rosetta structure prediction pipeline were used, instead of DMPfolds default method of generating an MSA from the UniClust30 database. All other parameters were kept to their default values. Details of the DMPfold model building can be found in Supplementary Methods section 2.

## Quality metrics: pairwise sequence identity, TM-score, cosine similarity

Model quality assessment and construction of the MIP curated are discussed in detail in Supplementary Note 3. The MQA scores for AlphaFold2 predictions are the mean pLDDT or pTM values. Pairwise sequence identity and structural similarity (TM-score) were generated using TM-align, which is a widely accepted tool in the community and is being continuously updated. Two structures were identified as similar (including novel fold identification) if the corresponding TM-score ≥ 0.5 (unless otherwise stated). We used DeepFRI for function prediction because it was specifically developed for this purpose and includes newest ML tools (such as GCNs and LSTMs), having been trained on current databases with hundreds of millions of data points. It further has the ability to generalize well, being able to predict any function across the GO tree. Detailed information on DeepFRI output, interpretation of scores and cosine similarity is provided in Supplementary Methods section 3. Pairwise functional similarity was computed as a cosine similarity between concatenated DeepFRI output vectors, which comprise scores for 6315 GO terms/EC numbers. Noise was reduced by only considering function scores above a threshold of 0.1.

## Protein universe visualization

For every structure in the MIP visualization dataset (comprising 10,000 Rosetta and corresponding DMPfold models plus 6,631 CATH 4.3.0 superfamily structures—see Tables S1 and S2), we generated a contact map (representing residues closer than <6Å from each other), which was then transformed into graph. Such graph representation was subsequently used to form a 42-dimensional graphlet vector[23]. The collection of graphlet vectors (26,631 ×42 matrix) was then projected onto a 3D space using two standard dimensionality reduction methods, i.e., UMAP and PCA. For UMAP we used the following set of parameters (which provided reasonable spread of the data with clear CATH class separability): n_neighbors = 100, min_dist = 0.001, N_components = 3, metric = cosine. Visualizations were created with Emperor[26]. An overview of the pipeline is depicted in Supplementary Fig. 49.

## Meta-data

MIP models were superimposed against all CATH 4.3.0 superfamilies (http://www.cathdb.info/, accession January 5, 2021) using TM-align in order to identify novel folds (see below) and annotate them using CATH classification, i.e., the most similar CATH structure to a given protein (with the highest TM-score normalized by MIP model size) is used as a template. The annotation quality drops with decreasing TM-score (which is important for novel folds) but the quality is high, especially at the class level. Proteins were annotated as α-helical transmembrane proteins if their OCTOPUS output contained at least one "M" segment. Similarly, proteins were annotated as β-barrel transmembrane proteins if their BOCTOPUS output contained at least eight "pL" segments. All the meta-data are discussed in Supplementary Note 4.

## Novel fold identification

To identify new folds, we started from high quality MIP models i.e. the *MIP_curated* dataset. First, we performed a TM-align structural superposition against CATH 4.3.0 superfamilies (see above). For the models without any significant structural similarity to CATH (TM-score <0.5), we performed a superposition against representative structures from the PDB90 (time stamp January 15, 2021). A putative novel fold is a high quality (i.e., from *MIP_curated*) single domain predicted by both Rosetta and DMPfold with satisfactory confidence (agreement TM-score ≥ 0.5 between Rosetta and DMPfold predictions) with a maximum TM-score <0.5 against CATH and the PDB90. Note that when comparing MIP and CATH/PDB structures we use the TM-score normalized by the MIP sequence length. The output set contained 452 structures grouped into 161 clusters. Clustering was achieved by computing pairwise TM-scores for all 452 models, then using the TM-scores as a list of edges and creating graphs using the NetworkX Python package. Node positions were computed using the Fruchterman-Reingold force-directed algorithm whereas connected components (clusters) were computed using the Breadth-first search algorithm. Both Rosetta and DMPFold datasets were clustered separately with the intersection of both sets being used as the final set of clusters. AlphaFold2 verification found 14 false positives, resulting in 438 novel structures grouped into 148

novel fold clusters. See Supplementary Note 5 for more information.

## DeepFRI prediction

DeepFRI computes saliency maps for each predicted GO term[6]. These maps identify residues that are important for this function but that does not mean that these are active residues, they could be important for protein stability or short sequence motifs away from the active site to identify the function of this protein. Heatmaps in Fig. 4 were generated based on curated MIP (Rosetta) models with DeepFRI score ≥ 0.2. Models were then grouped by GO-term and pairwise structural similarity was computed as the maximum TM-score of two superimposed MIP models (i.e. the larger of TM-scores normalized by the first and second sequence lengths was chosen).

## Reporting summary

Further information on research design is available in the Nature Portfolio Reporting Summary linked to this article.

## Data availability

All sequence, structure and function data generated in this study, along with relevant metadata have been deposited on Zenodo (url https://doi.org/10.5281/zenodo.6611431) and on Github at https://github.com/microbiome-immunity-project/protein_universe under commit ID 23354bf. This includes information on the directory structure and how to search the database via workflows and scripts using a query sequence, a query structure, or a query function, to find similar proteins in the MIP dataset. Source data are provided with this paper.

## Code availability

TM-align v.20190822 (https://zhanggroup.org//TM-align/) was used for computing TM-scores and sequence identities of aligned structures[12]. Structure visualizations were created in Pymol v.2.4.0 (https://github.com/schrodinger/pymol-open-source). Secondary structure assignments were generated using Stride v.20021022[46]. Alpha-helical transmembrane annotations were generated using OCTOPUS (as a part of TOPCONS2 software[47]; singularity image downloaded on July 17, 2020, dependencies: Blast v.2.2.26, Uniref90 v.20200119, Pfam 20191204). Beta-strand transmembrane annotations were generated using BOCTOPUS2[48] (zip downloaded on August 8, 2020; dependencies: HH-suite v.2.0.16, Blast v.2.2.26, Uniprot20 v.20160226). Absolute and relative contact order was computed from definitions[49]. For disordered sequence identification we used MobiDB-lite[50] v.1.0 (March 2016) and DISOPRED3[51] (zip downloaded on September 16, 2021; dependencies: Blast[52] v.2.2.26, Uniref90 v.20210731). Putative new fold clusters were computed using Python package NetworkX v.2.7.1. For putative new fold verification, we used AlphaFold2 with "preset" flag set to full_pdb (repository downloaded on August 16, 2021; reference databases which includes the PDB downloaded on July 31, 2021). A cosmetically modified version of the Rosetta Macromolecular Modeling Suite[8,53], based on release 2016.32.58837, was used for protein structure prediction on the World Community Grid. The fragment picking pipeline[54] is also part of the standard Rosetta distribution. Both are obtainable from the Rosetta Commons (https://www.rosettacommons.org/). Residue-residue pair constraints were obtained using GREMLIN[44] version 2.0.1. DMPfold[9] v1.0 (https://github.com/psipred/DMPfold, downloaded September 2019) was used to predict the structures of all MIP sequences. All custom codes generated for this study are part of the Zenodo repository (url https://doi.org/10.5281/zenodo.6611431) and on Github at https://github.com/microbiome-immunity-project/protein_universe under commit ID 23354bf. This includes information on the directory structure and how to search the database via workflows and scripts using a query sequence, a query structure, or a query function, to find similar proteins in the MIP dataset.

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

## Acknowledgements

We kindly acknowledge the support of the IBM World Community Grid team (Caitlin Larkin, Juan A Hindo, Al Seippel, Erika Tuttle, Jonathan D Armstrong, Kevin Reed, Ray Johnson, and Viktors Berstis), and the community of 790,000 volunteers who donated 140,661 computational years since Aug 2017 of their computer time over the course of the project. This research was also supported in part by PLGrid Infrastructure (to PS). The authors thank Hera Vlamakis and Damian Plichta from the Broad Institute for helpful discussions. The research reported in this publication was supported by the Flatiron Institute as part of the Simons Foundation to J.K.L., P.D.R., V.G., D.B., C.C., A.P., N.C., I.F., and R.B. This research was also supported by grants NAWA PPN/PPO/2018/1/00014 to P.S. and T.K., PLGrid to P.S., and NIH - DK043351 to T.V. and R.J.X.

## Author contributions

R.K., R.B., R.J.X., and T.K. conceived and initiated the project; T.K. and J.K.L. coordinated the project; P.D.R. and T.K. prepared and supervised World Community Grid project execution; V.G. developed the methodology and performed functional annotations; D.B. and P.S. developed, implemented and performed low-dimensional representation of the protein space; P.S., J.K.L., T.K., and P.D.R. analyzed the data; J.K.L., T.K., P.D.R., and P.S. wrote the manuscript with input from all authors; C.C. and P.D.R. prepared computational framework for data aggregation and analysis; T.K., P.D.R., J.K.L., B.C.T., and T.V. performed preliminary data analysis; S.J. and T.K. implemented sequence processing pipeline; A.P., I.F., and N.C. provided HPC cluster and computational support.

## Competing interests

R.B., V.G., and D.B. are currently working at Genentech and no explicit conflicts of interest result from this change in affiliation. All other authors declare no competing interests.
