## [Peer review file · Nature Communications]

REVIEWER COMMENTS

Reviewer #1 (Remarks to the Author):

This paper has the very laudable goal of expanding the structural and functional coverage of microbial genomes. However there are a number of significant concerns that preclude its publication in the present form. At the least, major revision is required.

1. While DMP fold is faster than Alphafold 2 (AF2), its performance is considerably worse especially for hard and very hard sequences. Since much is made that they have putatively discovered 148 novel folds, these are likely to be in the hard to very hard category where the performance of DMPfold and Rosetta is inferior to Alphafold 2. At the least, they should run AF2 on these sequences and see what it predicts. More generally, they really should just use AF2 for the 200,000 sequences and report the consensus of all three methods, as well as the highly confident structural predictions of from a single approach.

2. Do the Rosetta and DMPfold predictions agree for the 148 novel folds?

3. What confidence criteria is used to make a structural or functional prediction? How confident are the novel fold predictions? What is the precision of functional transfer between distant sequences. How different are the functions of the top scoring assignment and the next best one?

4. Please provide the histogram of the best TM-scores of the putative novel folds to full length structures in the current entire PDB. This will provide greater confidence that they are/are not novel. By the way, if the TM-score >0.4 the folds are basically the same. A Tm-score of 0.5 makes the assignment somewhat more confident. But suppose the score to an existing fold is 0.494, is it then a novel fold??

5. The continuity of structure space was pointed out in 2010. At the last they should refer to the paper by Skolnick and coworkers, see M. Gao and J. Skolnick. Structural space of protein–protein interfaces is degenerate, close to complete, and highly connected. Proc Natl Acad Science 2010: 107(52): 22517-22522. doi: 10.1073/pnas.1012820107.

Recommendation: This paper has a major strengths, namely the goal of increasing the structural/functional coverage of microbes but major revision is required if it is to achieve this objective.

Reviewer #2 (Remarks to the Author):

The current work presents a collection of novel structural models of bacterial proteins, not previously determined experimentally or computationally. The resulting effort (citizen science) and database of folds are important contributions to the state of the art. However, the analysis of this data presented in the manuscript is not sufficiently in-depth (for example, what are the functions of these new folds? Is the repertoire known and just the structures different? Are there new functions?), results/methods not well justified (see below, but also – every choice has consequences and should be addressed – e.g. BLAST vs MMSeqs for alignments, AlphaFold vs RosettaFold for folding, TM-align vs TopMatch for structural comparisons, GO-term distance/similarity, DeepFRI vs Vacic et al 2010 method, etc.), and, most importantly, the reported findings are not obviously significant or novel (see 30 years of structure/sequence/function work). The shortcomings of the analysis greatly dampen enthusiasm for this manuscript. See more comments below:

1. In Fig 2 caption "sequence identity correlates with structural similarity (a)" is not clear without more explicit explanations and numbers. There is a densely populated (red) set of alignments at sequence identity of 0-10% and structural similarity 0.2-0.4, which shows no correlation at all, just dots all over this space. What, I suppose, is meant here is more nuanced – for sequence similar proteins (say above 30% seq id), there is 'some' correlation (please give a number!) with structural similarity. I note that this observation is NOT unexpected and has been reported by many homology modeling and sequence analysis studies over the past 30 years (which should be cited, e.g. see Rost 1999 and on, the early Skolnick papers 2000 and on, and so forth).
2. The vast majority of methods are reported in the supplement. However, without the specifics of these methods, the results section is often meaningless; additionally, references to the specific method sections in the supplement are missing and are difficult to infer. As some, among many examples, consider these:
 - a. line 154 "By design, protein sequences in the MIP dataset are dissimilar (30% sequence identity cutoff)" is in conflict with referenced fig 2b, which clearly shows pairs over 30% seq ID. I suppose this observation is due to the domain similarity (not whole sequence), but more details are needed in text.
 - b. In the Methods paragraph of the paper it says that the authors "prioritized sequences according to their length and exhaustively sampled all putative novel domains between 40 and 200 residues" – this is a hard limit imposed by the authors on the sequences in the set (i.e. anything longer is excluded), so it therefore isn't clear why in the Results section the authors state "The sequence length distribution shows that our sequences are shorter than many of the proteins in the PDB, CATH or AlphaFold databases." Yes, they are shorter – you selected them as such
 - c. Visualization was created by generating a 42-dimensional graphlet vector representation (21) – requires an in-text brief explanation. Aside from an explanation of the method, some logic as to why this representation and dimensionality and methods were chosen are needed.
 - d. "For each of the 148 novel fold structural clusters" – how was the clustering done?

Please either summarize crucial pieces in main text or consider migrating some methods to main text.

3. It is not clear what the authors mean by “less biased protein structure universe”

4. Some set size explanations are needed. For example, why 3,052 in “We computed pairwise similarities between 3,052 random sequences in the curated dataset (MIP_random5000_curated - see Table S3)”. I note that the file name here says random 5,000, which suggests that originally 5,000 were chosen and some removed – why?

5. Cosine similarity ranges from -1 to 1, why are all scores here above 0? Is there a logic in comparing DeepFRI representations via cosine similarity?

6. This is a poor explanation of the observations in cluster I “Similar structures can achieve different functions due to the fact that the gene ontology database is organized in a hierarchical manner and that parent or child functions are related but still different.” First off, no data about how often this occurs in this set was presented. Second, using the hierarchical distance metrics from e.g. the CAFA experiment would eliminate the need for exact matching and give more weight to your words (Radivojac et al, 2013).

7. I disagree with “Figs 4a and b show two proteins that use the same structural motif for different functions” First off – this is in reference to figure 3 (not 4), I suppose. Second, I would like to see a structural similarity measure reported here (what is the TM-align score of these complete structures?). Finally, if the conversation is only about a functional tiny helix, highlighted in red in the figure, this comparison is equivalent to secondary structure comparison and can not be used to claim structural similarity. That is, many functions are carried out by tiny helices, and they change depending on sequence and surrounding structure....

On the same note, I am not sure that DeepFRI resolution is sufficient to make sweeping statements about shared functionality of different structures. Thus, in fig 3c, example 13 scores are fairly low (i.e. .21 and .15) as compared to the .1 cutoff introduced in text (gray in figures) – what does this mean as to accuracy of these predictions? Similar question holds for fig 3d, where values are .19 and .16

We very much thank the reviewers for their time, effort, and detailed feedback! We strongly believe that addressing these comments has elevated this manuscript to the next level. Detailed responses are below with reviewer's comments in **blue**, our brief responses in **black**, and text that was added or rewritten in **red**.

Reviewer #1 (Remarks to the Author):

This paper has the very laudable goal of expanding the structural and functional coverage of microbial genomes. However there are a number of significant concerns that preclude its publication in the present form. At the least, major revision is required.

1. While DMP fold is faster than Alphafold 2 (AF2), its performance is considerably worse especially for hard and very hard sequences. Since much is made that they have putatively discovered 148 novel folds, these are likely to be in the hard to very hard category where the performance of DMPfold and Rosetta is inferior to Alphafold 2. At the least, they should run AF2 on these sequences and see what it predicts. More generally, they really should just use AF2 for the 200,000 sequences and report the consensus of all three methods, as well as the highly confident structural predictions from a single approach.

2. Do the Rosetta and DMPfold predictions agree for the 148 novel folds?

We thank the reviewer for these comments and apologize for not making things sufficiently clear. We heavily edited the text and clarified the above points by adding two sections to the main manuscript:

Model quality assessment metrics to filter out low-quality models

Model quality assessment metrics were derived from 5,000 randomly selected proteins (a.k.a. *MIP_random5000_raw* - see Supplement section 5) in three steps. First, we noticed that the Rosetta models in our MIP database generally contain fewer coil residues than the DMPfold models (Supplement section 6.2.), yet the quality of the DMPfold models is higher for larger proteins (see Supplement section 6.3.). We therefore filtered by coil content with varying thresholds for the two methods: Rosetta models with >60% coil content, and DMPFold models with >80% coil content were filtered out as these have low quality.

Second, each modeling method needed a quality metric to evaluate the model quality. DMPFold outputs a confidence metric for each model that we used as-is. Rosetta's energy score is only meaningful in relation to other models of the same protein within the energy landscape. For Rosetta, we derived a model quality assessment (MQA) score by averaging the pairwise TM-scores¹⁰ of the 10 lowest-scoring models. If these models sample a minimum in the energy landscape, they are structurally very similar and their average TM-score is high. If Rosetta's scorefunction is unable to identify a specific fold, the average TM-score is low. We filtered out models with an MQA score ≤ 0.4 as these models have low quality.

The third quality metric we use is the agreement between the Rosetta and DMPFold models. If both models are similar (TM-score ≥ 0.5), then we can be confident that they are of high quality. This is supported by the correlation between MQA scores and TM-scores (Supplement section 6.3, Fig. S9). Further, the predictions between Rosetta and DMPfold mostly agree (mean TM-score = 0.61; median TM-score = 0.56) - Supplement section 8.3. The quality metrics we used are independent of target difficulty that is often used to classify targets in CASP. Further, all targets in our dataset are "hard" targets since they have low to very low homology to any known structures.

We then used these quality metrics to filter out low quality models: from >240,000 models in the *MIP_raw* database, we arrived at >200,000 models in the *MIP_curate* database (see Supplement section 5).

Identification and verification of novel folds

Putative new folds were identified by comparing our models against representative domains in CATH¹¹ and the PDB, using a TM-score cutoff^{12,13} of 0.5. The output set contained 452 novel structures grouped into 161 fold clusters. Putative novel folds were also verified by AlphaFold2, which identified 14 false positives, decreasing the number of novel structures to 438, clustered into 148 novel folds. Supplement section 8 describes in detail how novel folds were identified, why specific cutoffs were used and shows false positive clusters.

The agreement between AlphaFold2 and Rosetta or DMPFold is even higher than between Rosetta and DMPFold (see Fig. S50). We speculate that AlphaFold2 might link physical and knowledge-based scorefunctions (Rosetta) and machine learning approaches (DMPFold) better. We decided not to run AlphaFold2 on the entire MIP dataset due to runtime demands and us questioning whether this new data would provide much insight, as most of our Rosetta and DMPFold models have a high agreement and therefore high confidence.

3. What confidence criteria are used to make a structural or functional prediction? How confident are the novel fold predictions? What is the precision of functional transfer between distant sequences. How different are the functions of the top scoring assignment and the next best one?

For structure prediction, the confidence criteria are outlined in the answer to Q1 above. For novel fold identification, part of the answer is outlined in the Q1 above as well. We further re-ordered and edited the supplement to make things a lot more clear: We added a section in the supplement (Supplement section 4) on how DeepFRI works and how to interpret its output scores, which does not rely on homology transfer:

The confidence metric for DeepFRI should be interpreted in the following way: DeepFRI scores > 0.5 have a high confidence, which is mostly due to high occurrences in the training dataset and which often happens for more general functions at the top of the GO hierarchies. DeepFRI scores < 0.2 don't necessarily indicate an incorrect prediction, but rather that the number of examples in the training dataset is low. DeepFRI is further trained such that scores propagate

along the GO tree with higher scores towards the top of the tree and lower scores at the leaf, meaning that DeepFRI produces self-consistent predictions and does not "randomly" sample the GO tree. We often find that DeepFRI scores of 0.2 or even lower are perfectly adequate as indicated by very similar predictions for vastly different sequences (yet same structures), which is another indication of confidence in our predictions (see Fig. 4).

For functional similarity, we used cosine similarity between DeepFRI vectors thresholded at 0.1 (the non-zero threshold is important for denoising). DeepFRI was developed such that output scores are normalized between 0 and 1 and cosine similarity of two DeepFRI score vectors is a normalized sum of products of positive values (and therefore cannot be negative). We used cosine similarity as a similarity metric of two function prediction vectors because proteins have multiple functions and therefore DeepFRI predictions are vectors in a multi-dimensional function space. If two proteins have the same vectors, they would have the same function. If two proteins have the same functions but with slightly different scores, the function vectors would have a similar direction in that multi-dimensional space, therefore using cosine similarity as a similarity metric makes intuitive sense. In contrast, if two proteins have very different functions, the elements with non-zero scores in the two vectors would differ and therefore the cosine similarity would be zero or close to zero, meaning the directions of the score vectors in the multi-dimensional space would be different.

We further streamlined Supplement section 8 about novel fold identification, clarifying decisions about the cutoff used, showing all folds, showing false positives, and adding a section about the most-often seen functions for the novel folds (Supplement section 8.9):

8.1. Novel fold identification: overall procedure

Novel folds were identified in a two-step procedure: First, we compared protein models in the *MIP_curated* dataset to CATH superfamilies using the TM-score from TMalign. All resulting models without significant matches, i.e. that had a TM-score < 0.5 , were then compared against the PDB90, again using the TM-score from TMalign (see below for filtering procedure and cutoff justification). All models without matches in the PDB90 and which had a high agreement score between Rosetta and DMPFold models (agreement TM-score ≥ 0.5) were never-seen-before structures, of which were 452. These were then clustered into 161 novel folds (see below for clustering procedure) and verified through AlphaFold2, which identified 13 false positives, resulting in 438 novel structures clustered into 148 novel folds.

8.2. Novel fold identification: verifying a sensible TM-score cutoff

To choose and verify a sensible cutoff value for structural similarity using the TM-score, we computed the posterior probability that two folds within CATH 4.2.0 are similar (Fig. S36). It has been shown previously (see reference²⁹, Fig. 6) that a TM-score = 0.4 corresponds to at most 5% probability that two folds are similar.

Defining novel folds by a strict cut-off is debatable, but a 0.5 cut-off value is a meaningful choice given the data and is the most common definition used (see e.g. reference³⁰). For 0.55 the

probability of finding a known fold rises to 60-70% which still doesn't guarantee that a given fold is not novel. In these cases, a manual inspection of the models is required (see below).

Fig. S36: Validating the TM-score cutoff by computing the posterior probability that two proteins in CATH 4.2.0 share the same fold, verifying that a TM-score cutoff of 0.5 is a sensible choice.

8.3. Novel fold identification: comparison against CATH and PDB90

The CATH filtering step is depicted in Fig. S37 and shows maximum TM-scores for all *MIP_curated* models for both Rosetta and DMFold. Models that survived are placed in the lower left quadrant. Interestingly, the PDB filtering step is much more restrictive for Rosetta predictions - see Table S4. Coil content increases after each filtering step (see Fig. S39) but, in general, is still moderate. Such behavior is expected, since higher coil content generally translates into lower agreement with experimental (well-folded) structures deposited in CATH/PDB. The coil content is usually lower for Rosetta predictions which are constructed from fragments, in contrast to DMPFold. However, the coil content in our novel fold examples is relatively low (~30% on average).

8.9. Functions of the novel fold proteins

Fig. S58: The 50 most prevalent functions for the novel fold dataset, covering either the 148 representatives for all novel fold clusters (blue) or all 438 models (orange). For the histogram in the left panel, the function vectors were denoised with a score cutoff of 0.1, then summed over the models in the novel fold dataset and then sorted from the largest to the smallest scores. This means that the left histogram takes into account the magnitude of the scores. For the histogram on the right, the score vectors were binarized: scores ≤ 0.1 were set to 0 and scores > 0.1 were set to 1. Score vectors were summed over the models in the novel fold dataset and then sorted from the largest to the smallest scores. This means that the histogram on the right does not consider the magnitude of the scores but rather the prevalence of these particular functions.

4. Please provide the histogram of the best TM-scores of the putative novel folds to full length structures in the current entire PDB. This will provide greater confidence that they are/are not novel. By the way, if the TM-score >0.4 the folds are basically the same. A TM-score of 0.5 makes the assignment somewhat more confident. But suppose the score to an existing fold is 0.494, is it then a novel fold??

Thank you for this important request. Within the rewritten and clarified Supplement section 8 about novel fold identification, we added Figure S41 with the distribution of maximum TM-scores. Please refer to the supplement for the complete explanation.

8.5. AlphaFold2 verification of putative novel folds

We ran AlphaFold2 on all of 452 putative novel fold sequences. Fig. S40 compares Rosetta, DMPFold and AlphaFold2 predictions. The majority of our predictions agree well with AlphaFold2. Interestingly, the mean agreement TM-scores between Rosetta/AlphaFold2 and DMPFold/AlphaFold2 are higher than the one between Rosetta and DMPFold which may indicate that AlphaFold2 predictions possess some features of both Rosetta and DMPFold models.

Moreover, the agreement TM-score between Rosetta/AlphaFold2 and DMPFold/AlphaFold2 models correlates well with Rosetta/DMPFold MQA scores (see Fig. S42). We also see a slightly weaker correlation between the mean pLDDT score (i.e. AlphaFold2 MQA score) and Rosetta/DMPFold MQA scores (see Fig. S43).

Fig.

Fig. S40: Comparison between Rosetta, DMPFold and AlphaFold2 putative novel fold predictions. Points correspond to agreement TM-score between two given methods. Note the scale on the y-axis doesn't cover the full range of TM-scores from 0 to 1.

Fig. S41: Distribution of maximum TM-scores against PDB from putative novel folds dataset (see Table S5) for Rosetta and DMPFold (left) and AlphaFold predictions (right).

5. The continuity of structure space was pointed out in 2010. At the least they should refer to the paper by Skolnick and coworkers, see M. Gao and J. Skolnick. Structural space of protein-protein interfaces is degenerate, close to complete, and highly connected. *Proc Natl Acad Science* 2010: 107(52): 22517-22522. doi: 10.1073/pnas.1012820107.

We thank the reviewer for pointing this out and apologize for the oversight in citing the Skolnick paper. While we think that this particular paper isn't the best match because it talks about protein-protein interfaces, instead of protein folds, we have expanded on this discussion and added relevant papers, including several papers from Jeffrey Skolnick.

The microbial protein universe maps into a continuous fold space

[...]

The visualization further illustrates (see Supplement section 9) that the protein universe space is continuous, indicating that folds may evolve along a trajectory where small changes in the tertiary structure can eventually lead to a different fold. Our results are in agreement with prior work, albeit derived from a different (microbial proteins), larger and more diverse dataset and using a different methodology²⁷⁻²⁹. In contrast, a discrete fold space would display distinct clusters of folds that require larger conformational changes to interconvert between them. Prior work suggested evolutionary or geometric sources for the continuity of the protein fold space, and that a low-dimensional representation has the potential to aid protein structure-function investigations^{21,29-31}. We identify 438 previously unseen structures in our MIP dataset that cluster into 148 distinct, novel folds (46 clusters with multiple proteins and 102 singletons - see Supplement section 8). Fig 1d shows that the majority of novel folds are distributed throughout α/β fold space (compare with Fig 1c) with few novel folds in α or β fold space.

Recommendation: This paper has major strengths, namely the goal of increasing the structural/functional coverage of microbes but major revision is required if it is to achieve this objective.

Reviewer #2 (Remarks to the Author):

The current work presents a collection of novel structural models of bacterial proteins, not previously determined experimentally or computationally. The resulting effort (citizen science) and database of folds are important contributions to the state of the art. However, the analysis of this data presented in the manuscript is not sufficiently in-depth (for example, what are the functions of these new folds? Is the repertoire known and just the structures different? Are there new functions?), results/methods not well justified (see below, but also – every choice has consequences and should be addressed – e.g. BLAST vs MMSeqs for alignments, AlphaFold vs RosettaFold for folding, TM-align vs TopMatch for structural comparisons, GO-term distance/similarity, DeepFRI vs Vacic et al 2010 method, etc.), and, most importantly, the reported findings are not obviously significant or novel (see 30 years of structure/sequence/function work).

We thank the reviewer for these comments and questions. We want to provide a few responses here while also going into more detail in the specific questions below.

In the era of AlphaFold, we believe we need to rethink why structure matters and how we can leverage all the structural information. It is also fair to say that some of the facets of what we discussed in this manuscript here has been covered in the literature over the years, but it has never been analyzed so comprehensively on a novel dataset. Microbial proteins are understudied to date and we didn't know how their protein universe differs from the protein universe of known structures from a sequence-structure-function perspective. Further, what can we learn from such a dataset that we didn't already know? This is also the reason why we looked at the part of the dataset that doesn't follow the standard homology assumption that has fueled structural biology for the last 30 years. We strongly believe that we need new insights and ways of thinking about sequence and structure and we definitely need to include function in this picture if we want to make progress in our biological understanding of health and disease.

Looking at less explored corners of the protein universe highlights the importance of remote homology detection, a mindset of which we believe is insufficiently used to explore new biology. **As for functional annotations, the discovery of new functions requires new instance discovery which is an entire field of ML, which is not currently solved and is not part of this paper.**

As for justification of which tools were used and why: these choices were made based on the fit of the tool for the specific task in terms of sensitivity, runtime and technical hurdles. For instance, we used MMSeq for sequence alignments and clustering the initial sequences. AlphaFold wasn't available for the majority of the life of this project, which started in 2016, and computational runtime, code availability, and technical hurdles prevented us from running it on the World Community Grid. We used TM-align because it has been a widely accepted tool in the community, which is being continuously updated. We used DeepFRI for function prediction because it was specifically developed for the purpose of including newest ML tools (such as GCNs and LSTMs), having been trained on current databases with hundreds of millions of datapoints and being general to predict any function across the GO tree. These statements

unfortunately cannot be said about the function predictor from Vacic et al. which was developed in 2010, as neither the ML tools, nor training databases were anywhere close to what we have available now.

The shortcomings of the analysis greatly dampen enthusiasm for this manuscript. See more comments below:

1. In Fig 2 caption "sequence identity correlates with structural similarity (a)" is not clear without more explicit explanations and numbers. There is a densely populated (red) set of alignments at sequence identity of 0-10% and structural similarity 0.2-0.4, which shows no correlation at all, just dots all over this space. What, I suppose, is meant here is more nuanced – for sequence similar proteins (say above 30% seq id), there is 'some' correlation (please give a number!) with structural similarity. I note that this observation is NOT unexpected and has been reported by many homology modeling and sequence analysis studies over the past 30 years (which should be cited, e.g. see Rost 1999 and on, the early Skolnick papers 2000 and on, and so forth).

We apologize that we didn't sufficiently clarify this. Indeed, we wanted to confirm well-known relationships in context and make new observations for microbial proteins. We agree that the trend between sequence identity and structure similarity is not a mathematical correlation but rather follows a well-known trend. We have added Fig. S72 into the supplement and clarified this in the Figure caption and the text:

Figure caption for Fig. 2: [...] Analyses of these two datasets in this way lead us to the following conclusions: sequence identity and structural similarity follow a known trend (Fig. S72) [...]

10.1. Sequence-structure comparison

Fig. S72: Trend between sequence identity and structural similarity (using the TM-score) for the PDB baseline in Fig. 2a.

By design, the full-length protein sequences in the MIP dataset are dissimilar (30% sequence identity cutoff) yet pairwise sequence identities between domains can occasionally be higher than 30% (Fig 2). When correlating sequence identities to structural similarities for pairs of proteins, the vast majority of dissimilar sequences fold into different structures, and a trend can be identified (Supplement section 10.1.) that has been previously described^{32,33}. However, there are a fair number of proteins that have vastly different sequences and still fold into similar structures (Fig 2b). The PDB baseline that covers sequence similarities across all ranges from 0 - 100% confirms this expected trend, and it also confirms the general notion that similar sequences fold into similar structures (Fig 2a).

2. The vast majority of methods are reported in the supplement. However, without the specifics of these methods, the results section is often meaningless; additionally, references to the specific method sections in the supplement are missing and are difficult to infer. As some, among many examples, consider these:

We thank the reviewer for highlighting this. We agree that this was confusing – the writing style was an artifact from the word limit when we originally submitted this to *Nature*. We have rewritten the main text and added detail to the “online methods” section. We refer to the reviewer to the new version of the main manuscript.

a. line 154 “By design, protein sequences in the MIP dataset are dissimilar (30% sequence identity cutoff)” is in conflict with referenced fig 2b, which clearly shows pairs over 30% seq ID. I suppose this observation is due to the domain similarity (not whole sequence), but more details are needed in text.

We thank the reviewer for pointing this out. Correct, the discrepancy comes from the fact that the 30% sequence similarity threshold was put on full GEBA sequences (see “MIP dataset construction” section on p. 3 in the supplement) which were subsequently split into domains. We added a suitable explanation in the main text.

By design, the full-length protein sequences in the MIP dataset are dissimilar (30% sequence identity cutoff) yet pairwise sequence identities between domains can occasionally be higher than 30% (Fig 2).

b. In the Methods paragraph of the paper its says that the authors “prioritized sequences according to their length and exhaustively sampled all putative novel domains between 40 and 200 residues” – this is a hard limit imposed by the authors on the sequences in the set (i.e. anything longer is excluded), so it therefore isn’t clear why in the Results section the authors state “The sequence length distribution shows that our sequences are shorter than many of the proteins in the PDB, CATH or AlphaFold databases.” Yes, they are shorter – you selected them as such

Thank you for pointing out that this needs to be further clarified. Indeed, we focused on microbial structural domains that are generally shorter than eukaryotic ones [REF] with an average structural domain size is around 100 aa (Fig. 8 in

<https://www.ncbi.nlm.nih.gov/pmc/articles/PMC4256011/pdf/pcbi.1003926.pdf>). We have clarified this in the text.

Our MIP database is distinct from the other databases because it consists of proteins from Archaea and Bacteria, whose protein sequences are generally shorter than Eukaryotic^{15,16} ones. The average structural domain size for microbial proteins is about 100 residues (Fig 8 in reference¹⁷). We predicted structures in the size range from 40-200 residues because when we started this project in 2016, structure prediction methods performed better on smaller proteins, 200 residues still cover the majority of length space for microbial proteins, and we wanted to focus on single domains and longer proteins are more likely to cover multiple domains.

c. Visualization was created by generating a 42-dimensional graphlet vector representation (21) – requires an in-text brief explanation. Aside from an explanation of the method, some logic as to why this representation and dimensionality and methods were chosen are needed.

We agree and we have added an explanation to the text:

The microbial protein universe maps into a continuous fold space

We wanted to contextualize the MIP dataset in relation to existing structures and to investigate the features of a more complete and less biased protein structure universe^{18–21}. The PDB is biased by proteins that are more amenable to structure determination and by proteins of higher interest as pharmaceutical targets, resulting in a larger number of very similar structures with different mutations, ligands and chemical environments.

To generate the visualization, we represent each protein structure as a graph given by its C-alpha contact map below a 6Å threshold. We preprocess each graph by computing a 42-dimensional graphlet vector representation^{22,23}. A collection of graphlets up to size K is a set of all possible non-isomorphic induced at most K-sized subgraphs of a given graph G. Graphlet count vectors²⁴ report the counts of a set of (computationally tractable) graphlets up to a given size; they serve as a powerful baseline for graph encoding methods that do not consider node level features²⁵. For each model in the visualization dataset and CATH superfamilies, we mapped the 42-dimensional graphlet count vectors into 3D space using UMAP dimensionality reduction (Fig 1c and d). Visualization was done in Emperor²⁶. [...]

d. “For each of the 148 novel fold structural clusters” – how was the clustering done? Please either summarize crucial pieces in main text or consider migrating some methods to main text.

We agree that this information was hidden in the lengthy supplement. We have added a section to the Methods section:

The output set contained 452 structures grouped into 161 clusters. Clustering was achieved by computing pairwise TM-scores for all 452 models, then using the TM-scores as a list of edges

and creating graphs using the NetworkX Python package. Node positions were computed using the Fruchterman-Reingold force-directed algorithm whereas connected components (clusters) were computed using the Breadth-first search algorithm. Both Rosetta and DMPFold datasets were clustered separately with the intersection of both sets being used as the final set of clusters. AlphaFold2 verification found 14 false positives, resulting in 438 novel structures grouped into 148 novel fold clusters. See Supplement section 8 for more information.

3. *It is not clear what the authors mean by “less biased protein structure universe”*

Thanks for pointing this out. We have added a sentence:

The microbial protein universe maps into a continuous fold space

We wanted to contextualize the MIP dataset in relation to existing structures and to investigate the features of a more complete and less biased protein structure universe^{18–21}. The PDB is biased by proteins that are more amenable to structure determination and by proteins of higher interest as pharmaceutical targets, resulting in a larger number of very similar structures with different mutations, ligands and chemical environments.

4. *Some set size explanations are needed. For example, why 3,052 in “We computed pairwise similarities between 3,052 random sequences in the curated dataset (MIP_random5000_curated - see Table S3)”. I note that the file name here says random 5,000, which suggests that originally 5,000 were chosen and some removed – why?*

We apologize for creating confusion here. We have added a section to the supplement to clarify this and removed dataset sizes from this sentence in the main text.

5. MIP dataset descriptions

Datasets were created in the following order:

- (1) *MIP_raw* is the starting dataset with all entries, containing both Rosetta and DMPFold models of varying quality from both high-quality to low-quality models.
- (2) We used 5,000 random entries from the *MIP_raw* dataset to derive model quality metrics – this dataset is denoted *MIP_random5000_raw*.
- (3) When filtering out the low-quality models from *MIP_random5000_raw*, we are left with 3,052 high-quality models, denoting *MIP_random5000_curated*
- (4) When filtering out the low-quality models from the entire *MIP_raw* dataset, we get the *MIP_curated* dataset that only contains high-quality models.
- (5) We also created a subset of *MIP_curated* for visualization purposes, these are 10,000 pairs of high-quality Rosetta and DMPFold models.

The tables below describe our datasets and the number of models in each dataset. Note that there is partial overlap between the datasets of Rosetta models and DMPFold models because

either DMPFold didn't converge, or Rosetta models weren't completed due to interrupted runs on the World Community Grid.

Table S2: Definitions of the MIP datasets.

Dataset	Description
MIP_raw	All MIP entries. Includes both Rosetta and DMPFold models.
MIP_curated	Highest quality dataset created by filtering MIP_raw by quality metrics. Used for most quantitative studies in the main paper and the supplement.
MIP_visualization	Representative dataset from MIP_curated for visualization purposes; randomly sampled 9839 MIP IDs and added 161 representatives of novel fold clusters (in total: 10,000 points for DMPFold + 10,000 for Rosetta + 6631 CATH representatives).
MIP_random5000_raw	5000 random entries from MIP_raw with IDs common between Rosetta and DMPfold sampled by length distribution. The dataset was used to estimate the quality metric thresholds for constructing the MIP_curated dataset.
MIP_random5000_curated	The curated part of MIP random5000_raw was used to compute pairwise similarity measures in Fig. 2.

Table S3: Number of structures in the MIP datasets.

	Rosetta models	DMPFold models	CATH superfamilies	Common Rosetta-DMPfold models
MIP_raw	245,443	241,834	0	240,703
MIP_curated	211,069	203,877	0	184,642
MIP_visualization	10,000	10,000	6,631	10,000
MIP_random5000_raw	5000	5000	0	5000
MIP_random5000_curated	3052	3052	0	3052

5. Cosine similarity ranges from -1 to 1, why are all scores here above 0? Is there a logic in comparing DeepFRI representations via cosine similarity?

We agree that this wasn't sufficiently clear. We have added an explanation to the Supplement section 4:

For functional similarity, we used cosine similarity between DeepFRI vectors thresholded at 0.1 (the non-zero threshold is important for denoising). DeepFRI was developed such that output scores are normalized between 0 and 1 and cosine similarity of two DeepFRI score vectors is a normalized sum of products of positive values (and therefore cannot be negative). We used cosine similarity as a similarity metric of two function prediction vectors because proteins have multiple functions and therefore DeepFRI predictions are vectors in a multi-dimensional function space. If two proteins have the same vectors, they would have the same function. If two proteins have the same functions but with slightly different scores, the function vectors would have a similar direction in that multi-dimensional space, therefore using cosine similarity as a similarity metric makes intuitive sense. In contrast, if two proteins have very different functions, the elements with non-zero scores in the two vectors would differ and therefore the cosine similarity would be zero or close to zero, meaning the directions of the score vectors in the multi-dimensional space would be different.

6. This is a poor explanation of the observations in cluster I “Similar structures can achieve different functions due to the fact that the gene ontology database is organized in a hierarchical manner and that parent or child functions are related but still different.” First off, no data about how often this occurs in this set was presented. Second, using the hierarchical distance metrics from e.g. the CAFA experiment would eliminate the need for exact matching and give more weight to your words (Radivojac et al, 2013).

We thank the reviewer for pointing this out. After thinking about this more, we agree that this is a poor explanation as there is a disconnect between two things: The sentence mentioned above really speaks more to a functional similarity between two specific functions, which might be along the same branch in the GO hierarchy. In contrast, quadrant IV in Figure 2 (similar structures can have different functions) is for entire proteins, i.e. taking into consideration all possible functions of this protein and using cosine similarity as a comparison metric. **We have removed the sentence in question.**

As for the paper mentioned above: unless we missed it, the paper doesn't mention anything about word matching algorithms or hierarchical distance metrics. And as for language models that are able to extract meaning from words from the context, we highly doubt that a method from 9 years ago would be able to compete with current, more generalizable methods, as the machine learning and deep learning field evolves rapidly and latest models from Natural Language Processing are much more able to do just that.

7. I disagree with “Figs 4a and b show two proteins that use the same structural motif for different functions” First off – this is in reference to figure 3 (not 4), I suppose. Second, I would like to see a structural similarity measure reported here (what is the TM-align score of these complete structures?). Finally, if the conversation is only about a functional tiny helix, highlighted in red in the figure, this comparison is equivalent to secondary structure comparison and can not be used to claim structural similarity. That is, many functions are carried out by tiny helices, and they change depending on sequence and surrounding structure....

We corrected the mislabeling of the figure and apologize for not making this point sufficiently clear.

For the example in Fig. 3a the TM-scores are 0.55 and 0.70 for Rosetta / DMPFold models, respectively. For the example in Fig. 3B the TM-scores are 0.58 / 0.82 for Rosetta / DMPFold models, respectively. We respectfully disagree with the assessment that only a tiny helix is responsible for the function and that this relates to secondary structure instead of tertiary structure. The point we were trying to make is that the function is carried out by the same functional motif, which happens to be a small helix in Fig. 3A. Yet, the context in which this small helix exists, is the same in both proteins as the fold is the same. The same argument applies to the example in Fig. 3b where the N-terminus is responsible for the function in both cases. We hope that the similarities between both proteins' contact maps and Class Activation Maps from DeepFRI mapped into the sequence convinces the reviewer that the signal is not just in the secondary structure but indeed in the tertiary structure. We have added a section to the supplement to show the data and include the explanation.

11.1. Functional diversity of proteins with the same structure

Fig. 3 shows examples where we look at the functional diversity of proteins with the same structure. For the proteins in Fig. 3a the TM-scores are 0.55 and 0.70 for Rosetta / DMPFold models, respectively. For the proteins in Fig. 3b the TM-scores are 0.58 / 0.82 for Rosetta / DMPFold models, respectively. High similarities of the contact maps between the two proteins in each example indicate that it is the embedding of the structural motif (white rectangle) in the tertiary structure that is responsible for the function, instead of just the secondary structure. We also show the class activation maps with the functional predictions mapped onto the sequence to highlight the similarities there.

Fig. S78: Contact maps (left) and class activation maps (right) for both proteins in Fig. 3a. The boxes in white and black highlight the functional motif with the highest function prediction score.

Fig. S79: Contact maps (left) and class activation maps (right) for both proteins in Fig. 3b. The boxes in white and black highlight the functional motif with the highest function prediction score.

On the same note, I am not sure that DeepFRI resolution is sufficient to make sweeping statements about shared functionality of different structures. Thus, in fig 3c, example 13 scores are fairly low (i.e. .21 and .15) as compared to the .1 cutoff introduced in text (gray in figures) – what does this mean as to accuracy of these predictions? Similar question holds for fig 3d, where values are .19 and .16

We answered this question in point 3 to the first reviewer's comments. Further, the reviewer probably means confidence of the prediction instead of resolution? The resolution of DeepFRI predictions are on the residue level, which is significantly higher than for most tools available. The text that was added is in Supplement section 4:

The confidence metric for DeepFRI should be interpreted in the following way: DeepFRI scores > 0.5 have a high confidence, which is mostly due to high occurrences in the training dataset and which often happens for more general functions at the top of the GO hierarchies. DeepFRI scores < 0.2 don't necessarily indicate an incorrect prediction, but rather that the number of examples in the training dataset is low. DeepFRI is further trained such that scores propagate along the GO tree with higher scores towards the top of the tree and lower scores at the leaves, meaning that DeepFRI produces self-consistent predictions and does not "randomly" sample the GO tree. We often find that DeepFRI scores of 0.2 or even lower are perfectly adequate as indicated by very similar predictions for vastly different sequences (yet same structures), which is another indication of confidence in our predictions (see Fig. 4).

REVIEWERS' COMMENTS

Reviewer #1 (Remarks to the Author):

Overall, this is a significantly improved paper. There is one incorrect statement in the response that needs to be corrected.

The authors claim that a TM-score of 0.4 corresponds to a probability of the two folds being related is 0.05. This is wrong. Using eq 2 of the Xu/Zhang paper (see *Bioinformatics*. 2010 Apr 1; 26(7): 889–895.) with a TM-score of 0.4, the p-value is actually 3.54×10^{-5} , which means the probability that two folds are related at random is far less than 0.05. The paper and associated text needs to be corrected to reflect this.

Publish after this point is addressed.

Reviewer #2 (Remarks to the Author):

I appreciate the authors' efforts in replying to all raised questions and updating the manuscript. As is, it is a lot more understandable. I also reiterate that I believe the contributions of this work, and specifically the effort to increase the structural coverage of microbial space, are significant.

However, I am still not convinced by the interpretation of structural/sequence relationships with function. Chosen cutoffs (e.g. DeepFri at 0.2 is still a confident prediction?) contribute to the uncertainty. Having said that -- it is OK to disagree on interpretation of presented data, given that the data is freely available for everyone to make their own decisions.

One more minor comment, In caption of figure 2 -- "High sequence identity leads to high functional similarity (d), yet high functional similarity can be achieved by proteins with low sequence identity (d)." looks incorrect. In Figure 2d, either exceedingly few dots have "high" sequence identity (say >0.6) or there isn't an obvious relationship (space between 0.3 and 0.6 sequence identity is evenly populated in function scale). It might be that 2C should have been the first panel in that sentence... But even so, please make sure to describe what "high" id or similarity mean

We thank the reviewers for their time and effort in reviewing our manuscript again. Point-by-point responses are below with the reviewer's comment in **black**, our response in **blue**, and the text added to the supplement in **red**.

Reviewer #1 (Remarks to the Author):

Overall, this is a significantly improved paper. There is one incorrect statement in the response that needs to be corrected.

The authors claim that a TM-score of 0.4 corresponds to a probability of the two folds being related is 0.05. This is wrong. Using eq 2 of the Xu/Zhang paper (see Bioinformatics. 2010 Apr 1; 26(7): 889–895.) with a TM-score of 0.4, the p-value is actually 3.54×10^{-5} , which means the probability that two folds are related at random is far less than 0.05. The paper and associated text needs to be corrected to reflect this.

Publish after this point is addressed.

We thank the reviewer for highlighting this. The text in the supplement related to this point states

It has been shown previously (see reference 29, Fig. 6) that a TM-score = 0.4 corresponds to at most 5% probability that two folds are similar.

which includes the possibility that it may, in fact, be a lot smaller than 0.05. If we see this correctly, this reviewer's point actually corroborates our interpretation. To clarify this further, we have added the following sentence to the supplement page 28:

Further, according to Fig. 3 and equation 2 in the same paper (reference 29), the p-value for observing similar folds at random at a TM-score = 0.4 is 3.54×10^{-5} .

Reviewer #2 (Remarks to the Author):

I appreciate the authors' efforts in replying to all raised questions and updating the manuscript. As is, it is a lot more understandable. I also reiterate that I believe the contributions of this work, and specifically the effort to increase the structural coverage of microbial space, are significant.

However, I am still not convinced by the interpretation of structural/sequence relationships with function. Chosen cutoffs (e.g. DeepFri at 0.2 is still a confident prediction?) contribute to the uncertainty. Having said that -- it is OK to disagree on interpretation of presented data, given that the data is freely available for everyone to make their own decisions.

We thank the reviewer for pointing out that his/her scepticism stems from our claim that a DeepFRI prediction score of 0.2 can still be considered confident. We have added the following section to the supplement page 9:

The plot below shows the performance of DeepFRI on the independent test set that was used for evaluating the trained DeepFRI model - this test set had < 30% sequence identity to the training set. Here we evaluate the precision, recall, and F1-score for the function prediction vectors above a certain threshold, shown on the x-axis. For the molecular function branch of the GO-tree, a threshold of 0.2 gives an F1-value > 0.5 for the GCN (=GNN) which in our eyes is sufficiently confident for using this threshold for the prediction. We provide the scores here for the branches across the GO tree and Enzyme Commission number to allow anyone using DeepFRI to make their own decision on what prediction threshold they feel most comfortable with.

One more minor comment, In caption of figure 2 -- "High sequence identity leads to high functional similarity (d), yet high functional similarity can be achieved by proteins with low sequence identity (d)." looks incorrect. In Figure 2d, either exceedingly few dots have "high" sequence identity (say >0.6) or there isn't an obvious relationship (space between 0.3 and 0.6 sequence identity is evenly populated in function scale). It might be that 2C should have been the first panel in that sentence... But even so, please make sure to describe what "high" id or similarity mean

Thank you for pointing this out. The reviewer is correct - this is a typo and should have cited panel (c). We have also clarified this further.

High sequence identity (sequence identity > 50%) leads to high functional similarity (cosine similarity > 0.5) (c), yet high functional similarity can be achieved by proteins with low sequence identity (d).

The beginning of the caption already mentions which metrics we use for pairwise comparisons:

Pairwise comparisons of protein sequences (using sequence identity), structures (TM-score), and functions (cosine similarity between DeepFRI output vectors) for two datasets... .